# Design and Experiment of a Low-Loss Harvesting Test Platform for Cabbage

**Wenyu Tong** [1,2], **Jianfei Zhang** [1,*], **Guangqiao Cao** [1], **Zhiyu Song** [3,*] **and Xiaofeng Ning** [2]

1   Nanjing Institute of Agricultural Mechanization, Ministry of Agriculture and Rural Affairs, Nanjing 210014, China; tongwenyu@stu.syau.edu.cn (W.T.); caoguangqiao@caas.cn (G.C.)
2   College of Engineering, Shenyang Agricultural University, Shenyang 110866, China; ningxiaofeng@cass.cn
3   Graduate School, Chinese Academy of Agricultural Sciences, Beijing 100083, China
*   Correspondence: zhangjianfei@cass.cn (J.Z.); songzhiyu@cass.cn (Z.S.)

**Abstract:** In order to explore the mechanism and influence mechanism of cabbage harvest damage, a low-loss cabbage harvest test platform was designed on the basis of combining the physical characteristics of cabbage with the mechanical characteristics of mechanical harvest and the cabbage harvest operation process. Through the design of key components of the test platform harvesting, the key parameters of the pulling-out device, the reel device, the flexible clamping and conveying device, and the double-disc cutting device were determined. The movement changes of cabbage during pulling out, conveying, and cutting were analyzed to clarify the process of damage generation and critical conditions of damage in cabbage harvesting operations. The test results showed that when the speed of the pulling out device was controlled at 80–120 r/min, the speed of the clamping and conveying device was controlled at 120–240 r/min, and the speed of the double disc cutter was controlled at 140–180 r/min, the average success rate of pulling on the low-loss harvesting test platform was 92.7%; the average damage rate of the pulling process was 7.32%; the average success rate of clamping and conveying was 88.6%; the average damage rate of the clamping and conveying link was 12%; the average success rate of root cutting was 89.3%; and the average damage rate of the cutting link was 11.34%. The average qualified rate of harvesting in the pulling link was 86.7%, the average qualified rate of harvesting in the clamping and conveying link was 75.3%, and the average qualified rate of harvesting in the cutting link was 77.3%. All the performance indicators meet the design requirements and relevant standards, and the research results can provide a reference for the development and structural improvement of low-loss harvesting equipment for cabbage.

**Keywords:** cabbage; low-loss harvesting; mechanical harvesting characteristics; design of test platform

## 1. Introduction

Cabbage is one of the staple vegetables in China. It is planted in four seasons in the north and south of China [1]. It reaches 900,000 hm$^2$, and the total yield accounts for about 50% of the total yield of cabbage in the world [2]. At present, cabbage is still mainly harvested manually, and problems such as increased labor, increased labor intensity, and increased production costs have become increasingly prominent [3].

The soft and easily damaged characteristics of cabbage make their mechanized harvesting quality fluctuate during actual production work. The conveying mechanism structure of cabbage is mostly chain clamping [4], screw conveying [5], and clamping conveying [6]. The CKM-1 [7] and NKH-1 [8] single-row cabbage harvesters developed by the USSR adopt chain clamping and conveying, which have low harvesting efficiency and great damage. Hansen [9] from the United States applied for a patent for a cabbage harvester using a reverse rotation of the double helix conveyor and then transported it to the rear for cutting and packing; Bleinroth [10], Baker [11], and Mori G et al. [12] developed a cabbage harvester using a similar double helix conveying method. Cheng Zhou [13] and Dongdong Du [14] of

China improved and optimized the double helix conveying mechanism and increased the chain clamp mechanism to clamp the root of cabbage and cooperate with the double helix conveying to transport backward synchronously, but the test effect was still not satisfactory and the conveying damage was too large. In order to reduce the damage to cabbage during the conveying process, the conveyor belt conveying structure was invented and applied to the cabbage harvester [15]. Compared with screw conveying, clamping conveying can improve conveying stability and reduce cabbage damage by changing the clamping belt material and tensioning mechanism [16]. Lenker et al. [17] and Wadsworth et al. [18] used rubber as a conveyor belt to complete the transportation of cabbage. Bleinroth et al. [19] designed a pressure-top conveying structure based on the conveyor belt. When the conveying speed of the pressure-top conveyor belt is consistent with the pulling-out speed, the cabbage can be kept upright and pressed during the conveying process, which is conducive to the formation of lotus-type cabbage cutting and improves the accuracy of root cutting. In summary, the current research in the field of low-loss cabbage harvesting technology is relatively weak. The existing cabbage harvesting equipment is not suitable for China's planting agronomy, and there is still a problem of large harvesting damage. Integrating flexible clamping and conveying technology into the design of cabbage harvesting and conveying structures may be a breakthrough to solve the high damage rate in the cabbage harvesting process [20,21].

Therefore, this paper aims to improve the operational performance of the cabbage harvester and reduce harvesting damage. Cabbage was selected as the research object, and on the basis of clarifying the mechanical and physical parameters of cabbage, a low-loss harvesting test platform for cabbage was designed. By obtaining the critical conditions for the damage of cabbage in the harvesting process, the attitude migration law of cabbage in the harvesting mechanism was clarified, which provided a reference for the development and structural improvement of low-loss harvesting equipment for cabbage.

## 2. Analysis of Mechanical Harvesting Characteristics of Cabbage

The cross-section of the rhizome cutting of cabbage during the harvest period is composed of the central pith, xylem, and phloem fiber layers from inside to outside. The matrix leaves are thicker, the moisture content of the long-term exposed external leaves is lower than that of the internal leaves, and the toughness is stronger. The internal leaves have a high moisture content and are brittle. The basic physical properties and harvesting mechanical properties of cabbage were determined: the rhizome length, diameter, and pulling force were measured for cabbage. The universal testing machine was used to test the crushing, rhizome cutting force, water content, and cutting force of a single plant. The mechanical parameters such as root shear characteristics and ball crushing force under different water content conditions were tested, respectively, which provided a theoretical basis and data support for the study of cabbage harvesting equipment. See Figure 1.

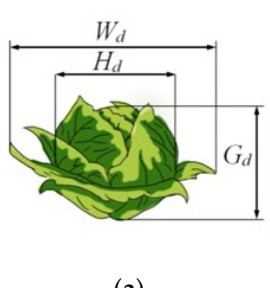 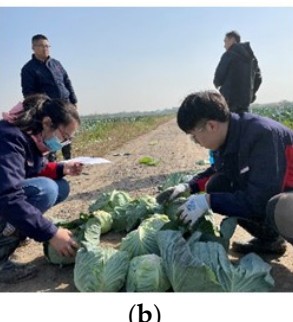 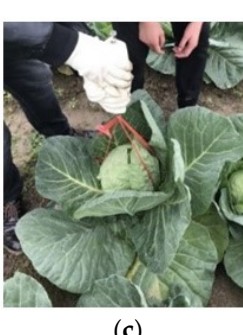 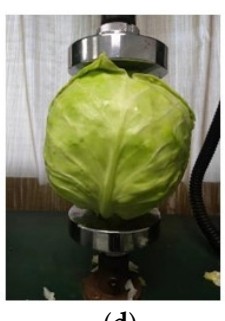 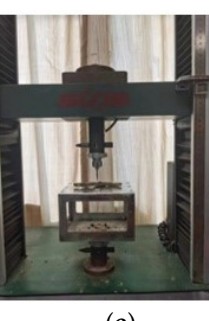

(**a**)          (**b**)          (**c**)          (**d**)          (**e**)

**Figure 1.** Determination of basic physical parameters and harvesting mechanical properties test of cabbage: (**a**) Physical characteristics acquisition: 1: $W_d$: Expansion degree 2: $H_d$: transverse diameter 3: $G_d$: vertical diameter (**b**) Physical characteristics acquisition process. (**c**) Determination of pull-out force. (**d**) Compression test. (**e**) Root cutting force test.

### 2.1. Measurement Results of Basic Physical Parameters of Cabbage

The determination results of the basic physical parameters of cabbage are shown in Table 1. The vertical diameter of the cabbage ball was (162.3 ± 38.16) mm, the transverse diameter of the sphere was (163.8 ± 25.84) mm, the weight of cabbage was (1.91 ± 0.68) kg, the diameter of the cabbage rhizome was (28.71 ± 4.56) mm, and the expansion degree of cabbage was (325.6 ± 40.89) mm. The measurement results can provide a theoretical basis for the spatial layout of the cabbage harvesting device.

**Table 1.** Results of basic physical parameters of cabbage.

| Statistical Indicators | Vertical Diameter of Cabbage (mm) | Transverse Diameter of Cabbage (mm) | Weight of Cabbage (kg) | Diameter of Rhizome (mm) | Expansion Degree of Cabbage (mm) |
|---|---|---|---|---|---|
| Average value | 162.3 | 163.8 | 1.91 | 28.71 | 325.6 |
| Maximum value | 190 | 185 | 2.265 | 31.8 | 360 |
| Minimum value | 135 | 140 | 1.241 | 24.6 | 290 |
| Standard deviation | 19.05 | 12.92 | 0.34 | 2.28 | 20.45 |
| Coefficient of variation | 0.12 | 0.08 | 0.18 | 0.08 | 0.06 |

### 2.2. Pulling Force Measurement Results

The determination results of cabbage pull-out force are shown in Table 2: The maximum pullout force of this variety was 228 N, the minimum was 154 N, and the average pullout force was 190.4 N. The measurement results can provide a theoretical basis for the design of the extraction device.

**Table 2.** Results of pulling force.

| Statistical Indicators | Pulling Force (N) |
|---|---|
| Average value | 190.4 |
| Maximum value | 228 |
| Minimum value | 154 |
| Standard deviation | 33.93 |
| Coefficient of variation | 0.18 |

### 2.3. Test Results of Rhizome Shear Force and Extrusion Force

It can be seen from Figure 2a,b that the curve of force and displacement in the test data is non-linear. During the shear test, the blade first contacts the outer epidermis of the cabbage rhizome, and the shear force gradually increases. When the blade cuts off the outer epidermis and enters the interior of the rhizome, the shear force will decrease, and the change range is small. When the blade cuts to the lower outer epidermis, the shear force gradually increases until the cabbage rhizome is completely broken. The reason for this is that the material of the outer epidermis of the cabbage is fiber. Compared with the internal matrix, its flexibility is much stronger, so the shear force is at its maximum when it just enters the fiber layer. The maximum compressional force is $F_{max}$ = 1198.3 N, and the maximum root cutting force is $F_C$ = 137.138 N. This value can provide the necessary parameter support for the cutting device and the clamping and conveying device of the cabbage harvesting device.

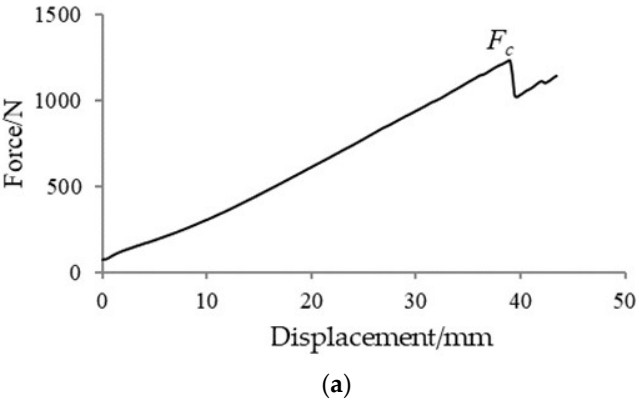
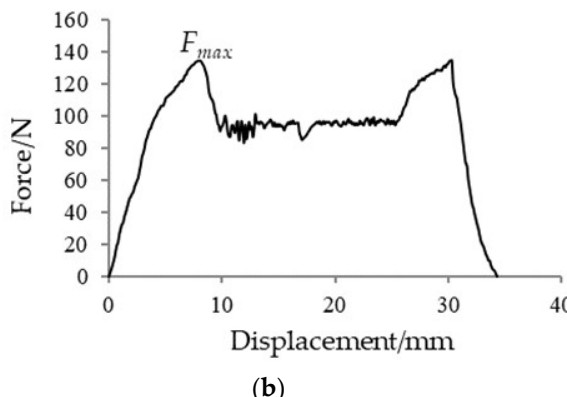

(**a**)　　　　　　　　　　　　　　(**b**)

**Figure 2.** Harvest mechanical properties test: (**a**) compressional force–displacement curve. (**b**) Root cutting force–displacement curve.

### 3. Machine Structure and Working Principles

#### 3.1. Structure of the Machine

The low-loss harvesting test platform for cabbage is mainly composed of a cabbage conveying system, cabbage harvesting system, test data acquisition system, and servo motor frequency conversion control box. The conveying system simulates the actual walking state of the cabbage harvesting machine in the field and relies on the cabbage harvesting system to simulate the real harvesting process. The specific structure of the prototype is shown in Figure 3. The conveying system for cabbage mainly consists of a cabbage rootstock clamping cup and conveying chain. The conveying speed (0~0.5 m/s) can be adjusted by the servo motor frequency control box. The cabbage harvesting system is mainly composed of a pulling device, reeling device, flexible clamping conveying device, and double disc cutting device. The speed of the pulling device (0~200 r/min), the speed of the reeling device (0~140 r/min), the speed of the flexible clamping conveyor belt (0~300 r/min), and the speed of the double disc cutting device (0~400 r/min) can also be adjusted by the servo motor frequency control box. Through the torque and pressure sensors installed in the harvesting device, the experimental data acquisition system can collect the motion parameters of the cabbage in the extraction, transportation, cutting, and other links, obtain the motion changes of the cabbage on the test header, find out the damage law, and determine the range of motion combination parameters between different key components. At the same time, the motion trajectory and speed-time curve of a single cabbage plant during transportation are calibrated and tracked. The damage to cabbage after transportation is recorded and saved, and performance indexes such as epidermal damage and cracking are measured.

#### 3.2. Principles of Test Platform

The operation process of the low-loss harvesting test platform for cabbage is shown in Figure 4. The test platform has preliminarily designed a "vertical clamping + flexible conveying" mechanism suitable for cabbage harvesting, which is mainly composed of a reel, a pulling roller, a flexible clamping belt, a cutterhead, a rack, and a transmission shaft. When the test platform is working, the pulling roller is placed under the outer leaf of the cabbages. The external rotation forces the root of the cabbages to be removed from the fixture to complete the pulling. The flexible conveyor belt clamps the cabbages and transports them backward. The cutter head cuts off the root of the cabbages and completes one test. By changing the material and structural parameters and motion parameters of the key components of the conveying mechanism, the damage to the cabbage after the conveying operation was recorded and saved, and the operation performance indexes such as epidermal damage and cracking ball were measured to find out the mechanism of harvesting damage to cabbage.

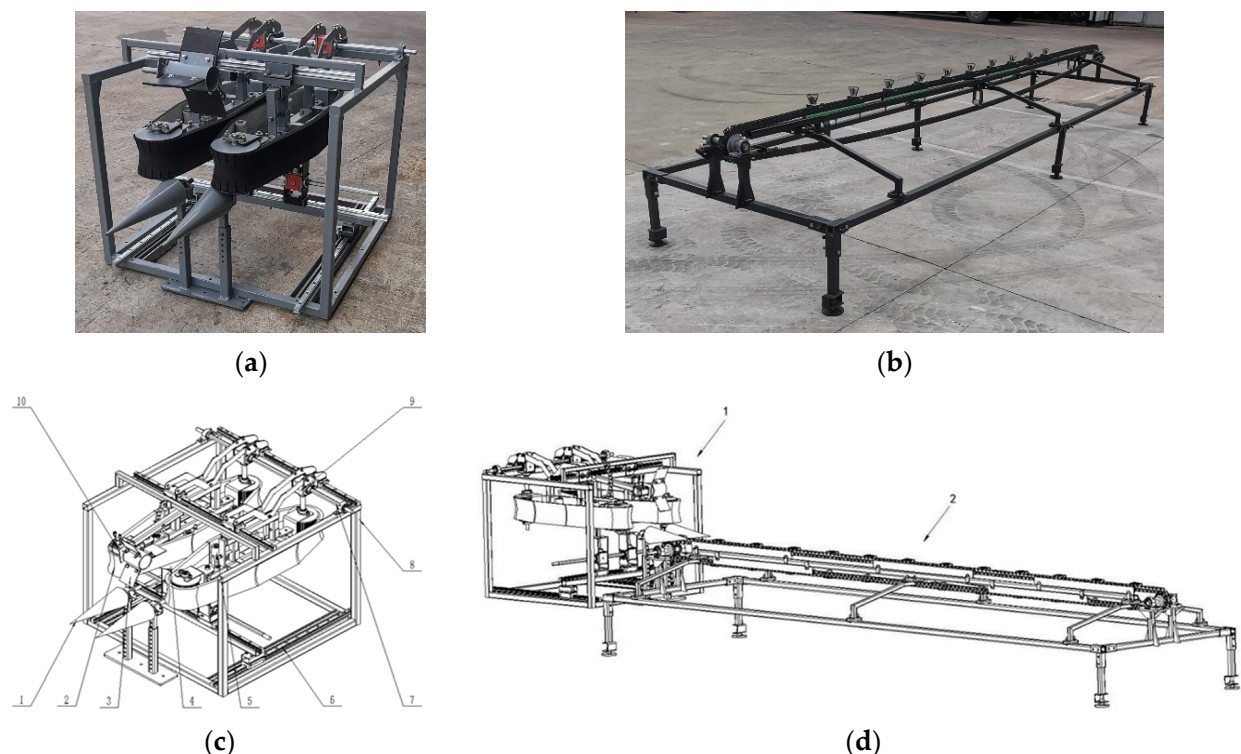

**Figure 3.** Structure diagram of low loss harvest test platform for cabbage: (**a**) Harvest cutting table. (**b**) Conveying test bench. (**c**) Harvest header structure diagram: 1: pulling device; 2: flexible clamping device; 3: pulling device; 4: disc cutter; 5: tensioning wheel; 6: track; 7: active shaft; 8: stand; 9: conveyor gear box; 10: pulling device motor. (**d**) Structure diagram of test platform: 1: harvest cutting table; 2: conveying test bench.

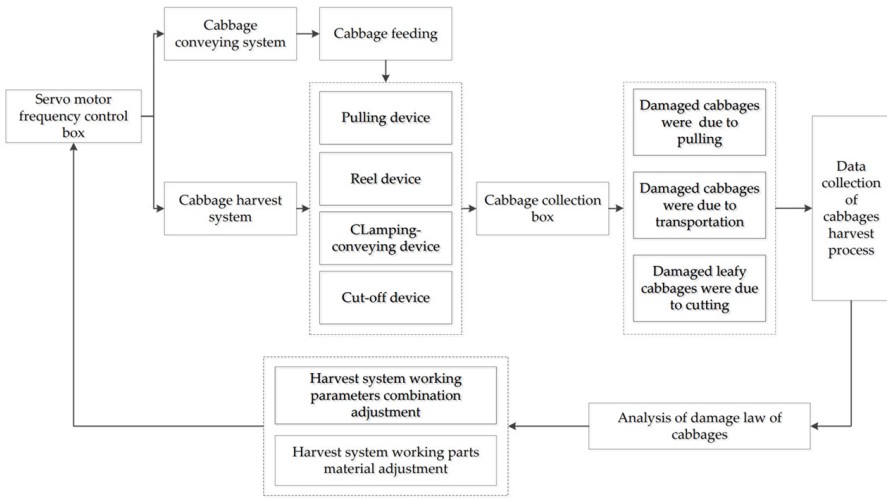

**Figure 4.** Test platform operation process.

When the low-loss harvesting test platform for cabbage works, firstly, the servo motor frequency conversion control box controls the working parts of the conveying test bench: the pulling device and the reeling device to run at a lower speed while keeping the clamping conveying device and the cutting device in a static state. At this time, the low-loss harvesting test platform for cabbage can only complete the pulling operation. Observe whether the conveying, pulling, and reeling links of the cabbage are smooth. Combined with the damage to the cabbages in the collection box and the data collection results of the harvesting process, the working parameters of the pulling device or the material of the

working parts are adjusted according to the test results until the conveying and pulling links are carried out smoothly and the cabbages are not damaged. Recording the working parameters (motion parameters and structural parameters) of the pulling device at this time is a critical condition for preventing damage to the cabbage in the pulling process. Similarly, according to this method, the servo motor frequency conversion control box controls the speed of the clamping and conveying device and the cutting device in turn, makes the low-loss harvesting test platform of the cabbage, completes the pulling + reeling + clamping and conveying, pulling + reeling + clamping and conveying + cutting links in turn, and records the critical conditions for damage in the clamping and conveying and cutting links, respectively.

*3.3. The Main Technical Parameters*

The main technical parameters of the low-loss harvesting test platform for cabbage are shown in Table 3.

**Table 3.** The main technical parameters.

| Parameters | Value |
|---|---|
| Harvest header size (length × width × height) (mm × mm × mm) | 1560 × 1300 × 975 |
| Conveying test bench size (length × width × height) (mm × mm × mm) (mm × mm × mm) | 5000 × 1300 × 780 |
| Conveying speed of conveying system/(m/s) | 0–0.5 |
| Pulling device speed/(r/min) | 0–200 |
| Reel speed/(r/min) | 0–140 |
| Clamping conveying device speed/(r/min) | 0–300 |
| Cutting device speed/(r/min) | 0–400 |
| Servo motor operating frequency/(Hz) | 0–50 |

## 4. Analysis of the Working Process and the Selection of Key Parameters

### 4.1. Design of Pulling Device

As shown in Figure 5a,b, the pulling device of the test platform is mainly composed of a reel and a pulling roller. When the pulling device works, the pulling roller is located below the cabbage. Through their own continuous external rotation, the cabbages are subjected to an upward pulling force. After the root of the cabbage is completely separated from the conveying system, it enters the clamping conveying mechanism through the right position of the reel above the pulling roller.

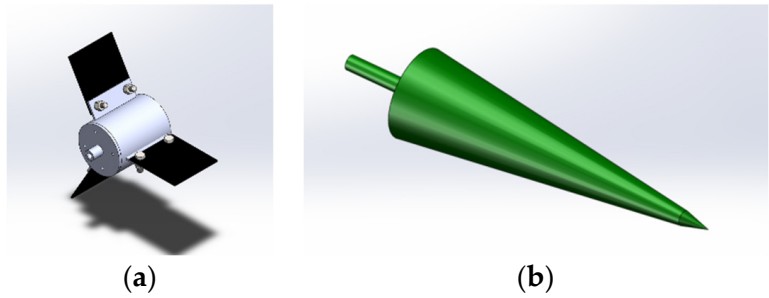

(**a**)　　　　　　　　　(**b**)

**Figure 5.** The pulling device: (**a**) reel; (**b**) pulling roller.

As shown in Formula (1), the ratio of the linear velocity at the outer edge of the reel to the forward speed of the machine is called the reeling speed ratio $\lambda$. When $\lambda \leq 1$, the cycloid amplitude of the working trajectory of the reel is small, and the function of supporting and guiding the cabbage cannot be realized. As shown in Figure 6, when $\lambda > 1$, the working trajectory of the reel is cycloidal. At this time, the reel can work normally, and the reel effect works well.

$$\frac{V_0}{V_x} = \lambda \tag{1}$$

where $V_0$ is the speed along the outer line when the wheel is working, m/s, and $V_x$ is the conveying speed of cabbage.

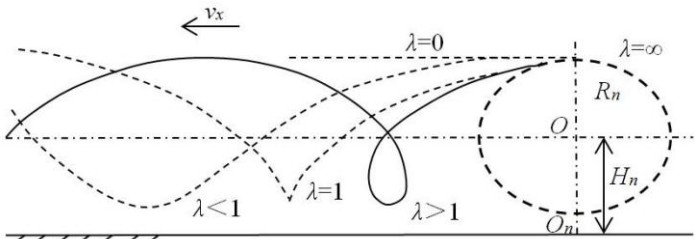

**Figure 6.** The movement track of the reel.

The displacement equation for the reel:

$$x = V_x + R_n \cos W_n t \tag{2}$$

$$y = H_n - R_n \sin W_n t \tag{3}$$

where $R_n$ is the radius of the reel, mm, and $W_n$ is the rotation speed of the reel, r/min. $H_n$ is the height from the center of the reel to the ground, mm.

It is assumed that the reel has "m" reel leaves. When a reel leaf rotates one circle, the forward distance of the harvester is:

$$L = V_x \frac{60}{mV_n} \tag{4}$$

where $L$ is the forward distance of the harvester, m, and $V_n$ is the angular velocity of the reel, rad/s.

When the reel works normally, the size of the reel should meet:

$$\frac{2\pi R_n}{m\lambda} > D. \tag{5}$$

where $D$ is the diameter of the cabbage.

In order to achieve continuous harvesting, the pitch of the long trochoid of the reel should meet:

$$S_n = \frac{2\pi R_n}{m\lambda} = \frac{S_l}{n} \tag{6}$$

where $S_n$ is the pitch of the long trochoid of the reel; $S_l$ is the distance between two adjacent cabbages; and $n$ is the reel leaf spacing, generally taking 1, 2, and 3.

The number of reel leaves on the test platform is 3, the radius of the reel is 240 mm, the distance of cabbage in the conveying system is 350 mm, take 1 for $n$, and the conveying speed of the conveying system is set to be 0.3 m/s. The rotation speed of the reel is set to 30, 50, or 70 r/min, and the trajectory of the reel is simulated by MATLAB-ADAMS. The simulation trajectory is shown in Figure 7.

*4.2. Design of Pulling Roller*

In order to analyze the movement of the cabbage on the pulling roller, the cabbage and the pulling roller are formed into a rigid body system for force analysis. In order to simplify the model, the position of the mass center is not considered in the analysis process. See Figure 8.

$$\sum m v_x = \sum F_x^{(e)}, \, m_j a_j - m(a_e - a_r \cos \delta) = F_N \cos \delta = \mu F_j \cos \delta \tag{7}$$

$$\sum m v_z = \sum F_z^{(e)}, \, F_m - (m_j + m)g - F_j \sin \delta = m a_r \sin \delta \tag{8}$$

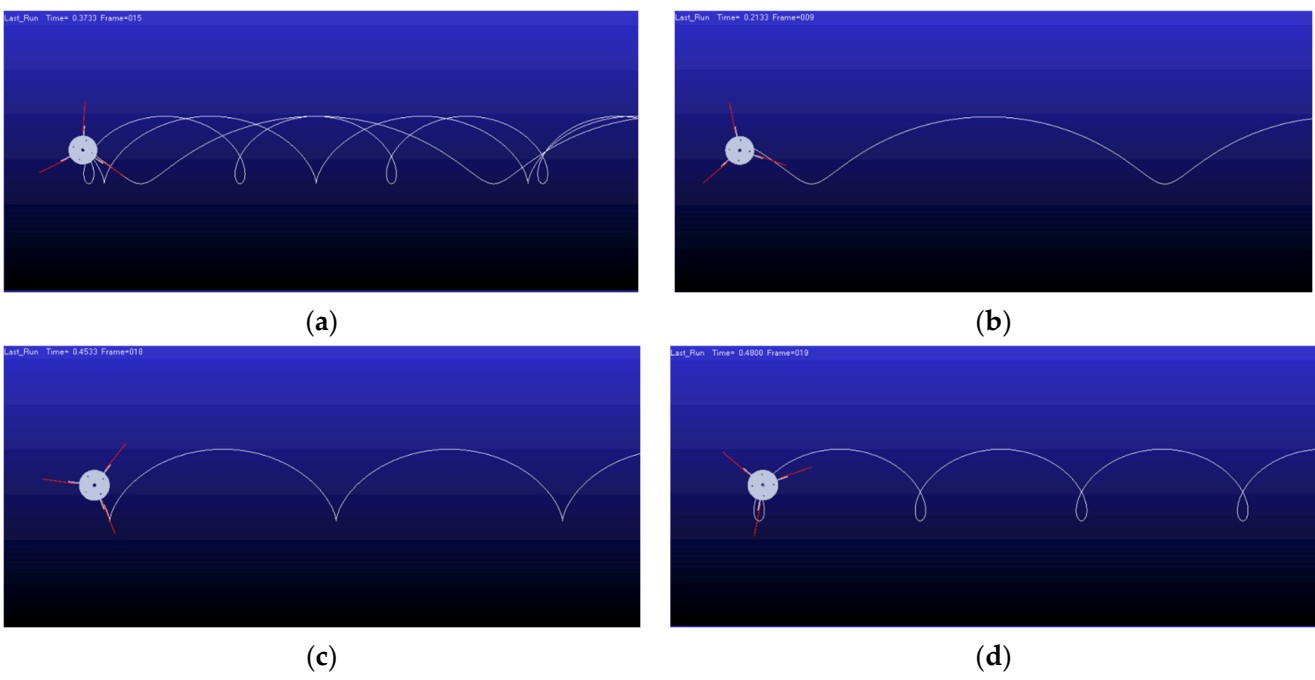

**Figure 7.** Trajectory diagram simulation of reel movement: (**a**) motion trajectory integration; (**b**) rotation speed = 30 r/min; (**c**) rotation speed = 50 r/min; (**d**) rotation speed = 70 r/min.

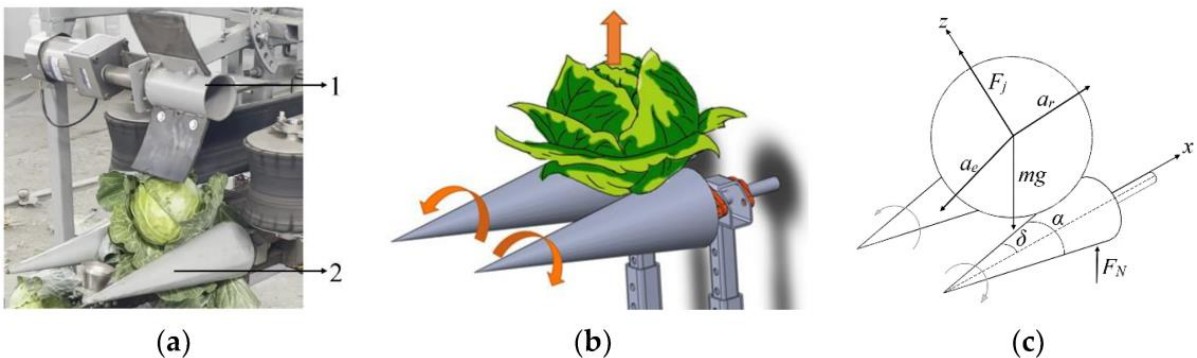

**Figure 8.** Dynamic analysis of pulling operation: (**a**) 1. Reel. 2. Pulling roller; (**b**) pulling roller and cabbage movement direction; (**c**) rotation speed = 50 r/min.

According to the centroid motion law of rigid body motion, the force balance equations in the *x*-direction and *z*-direction are listed:

$$m(a_j \cos \delta - \alpha_r) = mg \sin \delta \tag{9}$$

$$mg \cos \delta = F_j \tag{10}$$

From the above formula, it can be deduced that:

$$a_j = \frac{mg(\mu \cos^2 \delta + \sin \delta \cos \delta)}{m_j - m + m \cos^2 \delta} \tag{11}$$

$$a_j = \frac{g(\mu \cos^2 \delta + \sin \delta \cos \delta)}{K_M - 1 + \cos^2 \delta} \tag{12}$$

The conditions that must be met to make the cabbages move upwards without falling are: $\alpha_r > 0$. It can be derived from Formulas (8) and (12):

$$\frac{mg(\mu \cos^2 \delta + \sin \delta \cos \delta)}{m_j - m + m \cos^2 \delta} > g\frac{\sin \delta}{\sin \delta} \tag{13}$$

$$\mu > \frac{\tan \delta (K_M - 1 + \cos^2 \delta)}{\cos^2 \delta} - \tan \delta \tag{14}$$

where $\mu$ is the friction coefficient of cabbage; $F_m$ is the overall force of the pulling roller on the cabbage, N; $\alpha_r$ is the relative acceleration of the motion of the cabbage, in m/s$^2$; $\alpha_e$ is the acceleration of the conveying system, in m/s$^2$; $\delta$ is the angle between the cone on the pulling roller and the horizontal line, (°); $F_j$ is the pull-out force of the pull-out roller on cabbage, N; $\alpha_j$ is the absolute acceleration of the conveying system, in m/s$^2$; $m_j$ is the weight of a single pulling roller, in kg; $F_n$ is the friction force between cabbage and the pulling roller, N; and $K_m$ is the mass ratio of the pulling roller with cabbage.

To ensure that the cabbages do not fall during harvest, it is necessary to meet Formula (14). Therefore, the material properties should be considered when designing the pulling roller. It can be seen from Formula (12) that the relative motion between the pulling roller and the cabbage has a great influence on transportation, so the rotation speed of the pulling roller plays a vital role in the qualified rate of harvesting.

As shown in Figure 9, the force analysis of cabbage in the pulling process is carried out.

$$f_1 \cos \alpha = F_{N1} \sin \alpha \tag{15}$$

$$f_1 \sin \alpha + F_{N1} \cos \alpha = mg \tag{16}$$

$$f_1 = \mu F_{N1} \tag{17}$$

where $f_1$ is the resistance and friction of the cabbage in the pulling process, N; $F_{N1}$ is the supporting force of the working surface of the pulling roller on the cabbage, N; and $\alpha$ is the pulling angle of the pulling roller, (°).

When the cabbage is pulled out, it is subjected to resistance, friction, and support. After overcoming gravity, the pull-out force of cabbage is:

$$F = f_1 \sin \alpha + F_{N1} \cos \alpha - mg \tag{18}$$

It can be derived from Formulas (15)–(17):

$$F = \frac{mg}{\tan \alpha \sin \alpha}(\mu \sin \alpha + \cos \alpha) \tag{19}$$

The weight of cabbage is determined by Table 1 at 1.91 kg. According to Formula (19), the pulling force $F$ of the pulling roller on cabbage depends on the pulling angle $\alpha$. Therefore, a better pulling effect can be obtained by selecting the appropriate pulling angle. The material of the pulling roller designed in this paper is stainless steel, the friction coefficient is 0.3, the distance between the inside of the pulling roller is 50–80 mm, the diameter of the end is 120 mm, the total length of the conical pulling roller is 450 mm, and the angle between the pulling roller (taper) and the horizontal angle is 13.9°. At this time, the pulling force of the two pull-out rollers on cabbage is:

$$2F = \frac{1.91 \times 9.8}{\tan 13.9° \sin 13.9°} \times (0.3 \sin 13.9° + \cos 13.9°) = 677 \tag{20}$$

At this time, the theoretical pulling force of the pulling roller designed in this paper is greater than the pulling force measured in Table 2, which meets the design requirements.

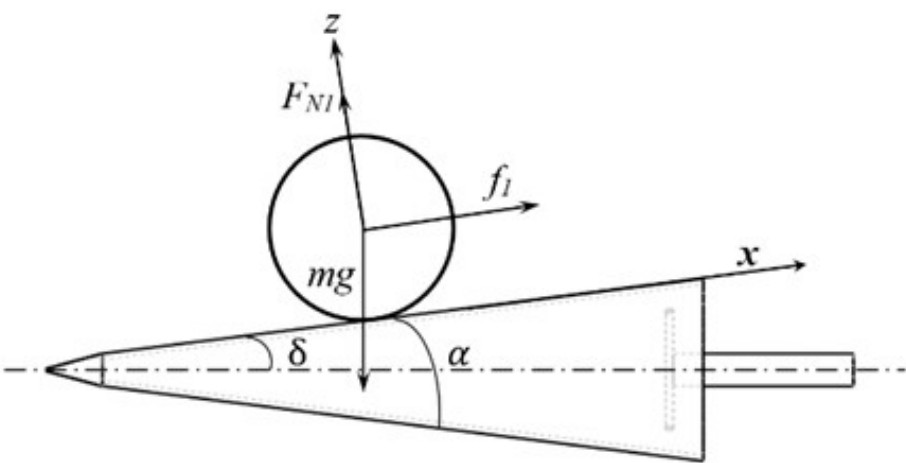

**Figure 9.** Pulling force analysis.

### 4.3. Design of Clamping Conveying Device

As shown in Figure 9, the low-loss harvesting test platform for cabbage designed in this paper adopts a new mechanism and new method called "vertical clamping + flexible conveying". By using the flexible feeding and flexible clamping methods, we can improve the adaptability of different ball diameters of cabbage and realize low-loss transportation [22,23].

The analysis of the movement process of the cabbage in the clamping and conveying device is shown in Figure 10, where 1 is the cabbage feeding link, 2 is the clamping and conveying link, and 3 is the harvesting of the finished product link.

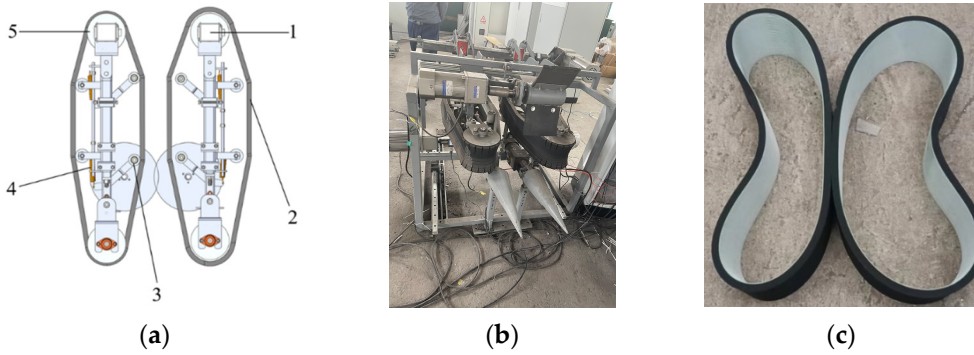

| (**a**) | (**b**) | (**c**) |

**Figure 10.** Clamping and conveying device: (**a**) 1: driving motor; 2: CR flexible sponge conveyor belt; 3: tensioner pulley; 4: tightening spring; 5: conveyor belt drive wheel. (**b**) Clamping conveying device physical diagram n. (**c**) High-density CR flexible sponge.

The clamping conveying process of cabbage is shown in Figure 11.

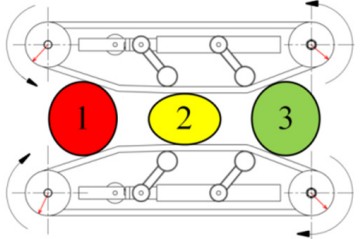

**Figure 11.** Clamping conveying process of cabbage.

The motion analysis and force analysis of the feeding link of the cabbage are shown in Figure 12a,b.

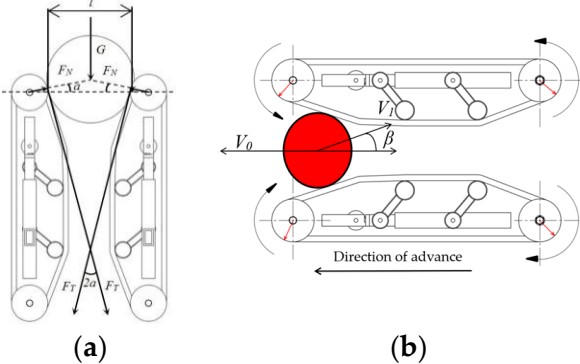

**Figure 12.** Clamping conveying process of cabbage. (**a**) Force analysis of the feeding link. (**b**) Motion analysis of the feeding link.

If the cabbages are not blocked in the feeding link and smoothly enter the clamping and conveying link, the following formula should be satisfied:

$$\tan \alpha \leq \frac{F_T}{F_N} = \mu \tag{21}$$

$$V_1 \sin \beta > V_0 \tag{22}$$

According to the above formula, it can be calculated by the following formula:

$$\mu \geq \sqrt{\frac{(D+d)^2 - (D-l)^2}{(D+l)^2}} \tag{23}$$

where $\alpha$ is the angle between pressure and the horizontal line of cabbage; $V_1$ is the linear velocity of the conveyor belt, m/s; $\beta$ is the lifting angle of the conveyor belt, (°); $V_0$ is the operating speed of the conveying system, m/s; $F_T$ is the friction force on cabbage, N; $F_N$ is the pressure of the conveyor belt on cabbages, N; $\mu$ is the friction coefficient between the conveyor belt and cabbages; $D$ is the diameter of the conveyor belt drive wheel, mm; $d$ is the diameter of cabbage, mm; and $l$ is the distance between the two conveyor belt drive wheels, mm.

The feeding inlet clamping position of the conveyor belt should be at the waist of the cabbage. In this paper, the single weight of "Chun xi" cabbage was 1.2–1.5 kg, the bulb was 180–200 mm, the feeding inlet spacing of the clamping conveying mechanism, which can be adjusted by spring and the minimum spacing, was 120 mm, and the diameter of the conveyor belt drive wheel was 110 mm. Therefore, the maximum value of the friction coefficient between the conveyor belt and the cabbage can be calculated as follows:

$$\mu \geq \sqrt{\frac{(110+200)^2 - (110+120)^2}{(110+120)^2}} = 0.9 \tag{24}$$

The maximum extrusion force of cabbage is:

$$F_{Nmax} = \frac{G}{\mu} \times 2 = 130.67 \tag{25}$$

According to the test of the mechanical harvesting characteristics of the cabbage, the extrusion force is far less than the maximum extrusion crushing force of 1198.4 N. Based on the compressional force test results, the clamping and conveying mechanical structure designed in this paper can avoid the results that the cabbage cannot be clamped due to too small extrusion pressure, and if the extrusion pressure is continuously increased, the

cabbage may be blocked, so the structural parameters of the clamping and conveying device are designed reasonably.

By analyzing the force on the base surface of the conveying interval and the deformation of the cabbage [24], the deformation analysis in the extrusion deformation link is shown in Figure 13.

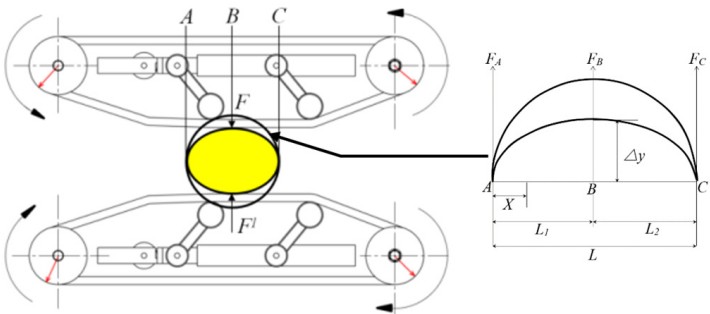

**Figure 13.** Deformation analysis of cabbage.

It can be seen from Figure 12 that the extrusion force $F = F_1$ at point B and the forces $F_A$ and $F_C$ at points $A$ and $C$ are:

$$F_A = F_C = \frac{FL_1}{L} \tag{26}$$

The reaction force of point $B$ can be obtained from the equilibrium equation:

$$F_B = \frac{FL_2}{L} \tag{27}$$

The deformation bending moment of cabbage is:

$$M_x = \frac{FL_2}{L}x - F(x - L_1) \quad (L_1 \leq x \leq L) \tag{28}$$

The deformation-bending moment is the same on both sides, and the integral on one side can be obtained:

$$EJ\frac{d^2y}{dx^2} = \frac{FL_2}{L}x \tag{29}$$

The deflection curve equation is:

$$y = \frac{F_4L_2x}{6LEJ} = x^2 - L_2^2 + L^2 \tag{30}$$

Because $x = L_1 = L_2$, the clamping deformation $\Delta y$ of cabbage is:

$$\Delta y = \frac{F_2L_2L_1L}{6EJ} \tag{31}$$

where $E$ is the modulus of elasticity; $J$ is the momentum of inertia; $L$ is the length of $AC$, mm; $L_1$ is the length of $AB$, mm; and $L_2$ is the length of $BC$, mm.

According to Formula (15), when the conveyor belt speed is too low, the conveying efficiency will be reduced, which makes it easy to cause conveying blockage, resulting in unsmooth subsequent operations and incomplete root cutting. Through the preliminary test and observation, the main forms of damage in the transportation of cabbages are friction, extrusion, collision, and other forms of damage. The reason is that the deformation of cabbage is too large under the action of rigid parts in the conveying process. If the deformation deflection $\Delta y$ of cabbage is too large, it is easy to squeeze and break. In Formula (25), the deformation deflection of the cabbage is determined by the elastic modulus. In order to reduce the damage as much as possible, the high-density CR flexible sponge belt is selected

to wrap the cabbage for clamping and conveying. While reducing the relative friction, the tensioning mechanism transfers part of the deformation of the cabbage to the flexible clamping conveyor belt. It has a certain anti-deformation effect on the cabbages and can ensure that the cabbages do not slide when they are clamped by the conveyor belt and will not cause ineffective root cutting of the cabbage due to sliding.

In order to adapt to different kinds of cabbage and improve the adaptability of harvesting equipment. The maximum center spacing of the conveyor belt designed in this paper can be adjusted to 280 mm, and the feeding inlet spacing can be adjusted in the range of 200–250 mm. Ensure that different varieties of cabbage can be successfully clamped and transported, even if the ball diameter is different.

### 4.4. Design of Root Cutting Device

The root-cutting device is one of the key components of the cabbage harvester. It works together with the clamping and conveying device. The main function is to cut off the root of the cabbage. The cutting effect has a great influence on the quality and efficiency of the subsequent harvesting operation. As the rhizomes of cabbage are relatively thick and some kinds of rhizomes are severely fibrotic, when a single disc cutter is used, a higher speed is required to reduce the imbalance of cutting force, which will cause considerable power consumption. Therefore, in order to ensure the stability of the force when cutting the root of the head cabbages and avoid incomplete cutting of the root of the head cabbages, this paper adopts the double disc cutting form. The two cutters keep a certain distance in the direction of the center line, and the two cutters should overlap a little to balance the horizontal force on the rhizome of the cabbage during the root-cutting process, so as to ensure the flatness and integrity of the root cutting. See Figure 14.

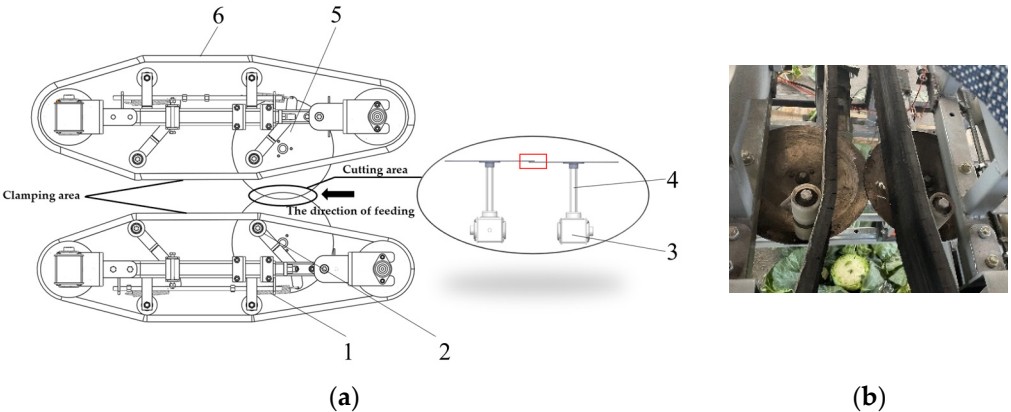

(**a**)            (**b**)

**Figure 14.** Root cutting device: (**a**) 1: tightening spring; 2: tensioning Wheel; 3: driving motor; 4: transmission shaft; 5: disc cutter; 6: conveyor belt. (**b**) Cutting device physical picture.

As shown in Figure 15, the power of the root-cutting device comes from two servo motors. In the process of root cutting, the roots of cabbage are subjected to cutting forces $F_1$, $F_2$, and $F_3$ in the $X$, $Y$, and $Z$-directions; $F_{1xy}$ and $F_{3xy}$ are the projection components of $F_1$ and $F_3$ in the $XY$ plane, respectively; and $F_{1x}$ and $F_{1y}$ are the projection components of $F_{1xy}$ in the $X$-axis and $Y$-axis, respectively. $F_{3x}$ and $F_{3y}$ are the projection components of $F_{3xy}$ in the $X$-axis and $Y$-axis, respectively. $\beta$ and $\varphi$ are the angles between $F_{1xy}$ and $F_{3xy}$ and the $X$-direction, respectively. Because the cutting positions of the left and right cutters are asymmetric, and the two cutters overlap in the axial direction, there is still some cutting force in the $X$-axis and $Z$-axis directions ($F_2$). Under the combined action of the forces $F_{1z}$ and $F_{3z}$ in the $Z$-axis direction and the feeding direction and the cutting friction force of the cabbage, the longitudinal component force above the cabbage is larger after receiving, which is helpful to clamp the rhizome of the cabbage [25]. Due to the actual root-cutting process, the vertical plane angle $\gamma$, which is the angle between the resultant force $F_{3xy}$ and the force $F_3$ in the $XY$ plane, is small; therefore, $F_{1x}$ can be approximated as the root main

cutting force. In addition, considering the actual installation and use, $\gamma$ is set to zero, and only the influence of pitch angle $\theta$ (the angle between the resultant force $F_{1xy}$ and the force $F_1$ in the $XY$ plane) is considered in order to determine the best cutting parameters.

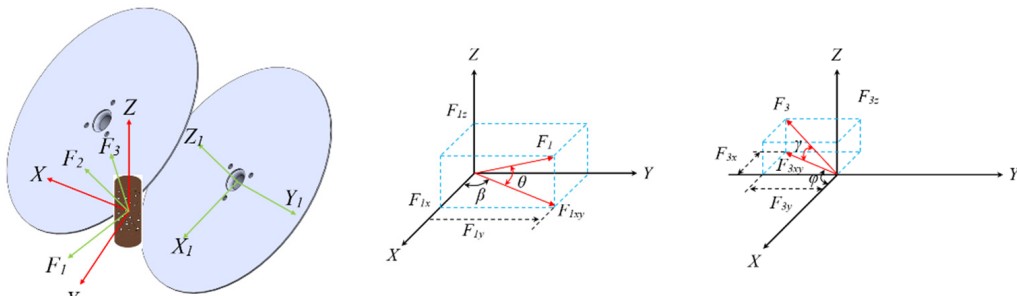

**Figure 15.** Root cutting force analysis.

As shown in Figure 16, it is assumed that the two disc cutters of the root cutting device are ideal discs, the cabbage is idealized as a round, and the diameter of the rhizome is $D_1$.

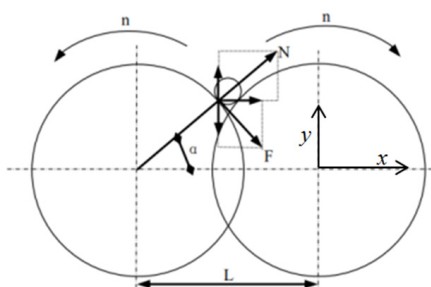

**Figure 16.** Force on the rootstock of cabbage.

From the force analysis in the diagram, the following equations can be obtained:

$$R_X = N_X + F_X \tag{32}$$

$$T_Y = F_Y + N_Y \tag{33}$$

where $R_X$ is the root cutting force and $T_Y$ is the clamping force of the cutter on the rhizome. $N$ is the normal reaction force of the disc cutter on rhizomes; its horizontal component is $N_X$ and its vertical component is $N_Y$, N; $F$ is the friction of the circular cutter disc on the friction force of the circular cutter on the root of cabbage; its horizontal component is $F_X$ and its vertical component is $F_Y$, N.

In order to make the rhizome be held by the disc cutter, the following conditions must be met [26]:

$$T_Y > 0 \tag{34}$$

It can be deduced that: $F_Y > N_Y$, where $F = N \cdot f$, that is:

$$N \cdot f \cos \alpha > N \cdot \sin \alpha \tag{35}$$

Therefore, when $f > tan\alpha$, the disc cutter has better clamping performance:

$$\alpha = \cos^{-1} \frac{L/2}{(D_1 + D_2)/2} = \cos^{-1} \frac{L}{D_1 + D_2} \tag{36}$$

where $f$ is the friction coefficient between the disc cutter and the rhizome of cabbage; $\alpha$ is the angle between the reaction force of the disc cutter on the cabbage rhizome and the

$x$-axis, (°); $L$ is the distance between two disc cutters, mm; $D_1$ is the root diameter of the cutting point, mm; and $D_2$ is the diameter of the disc cutter, mm.

The center distance of the designed double disc cutter is 190 mm, the diameter of the disc cutter is 200 mm, the angle $\alpha = 31.02°$, and the overlap thickness of the two cutters is 5 mm. At this time, $f > tan\alpha$, which can meet the requirements of clamping performance.

### 4.5. Design of Data Acquisition System

By installing torque and pressure sensors on the test platform to collect the speed and displacement of the cabbage in the harvesting system, the motion trajectory and speed–time curve of a single plant during the harvesting process are calibrated and tracked, and the damage to the cabbage after the harvesting operation is recorded and saved in order to find out the results of the movement of the cabbage in the pulling, conveying, and cutting of roots and determine the range of working parameters between different harvesting components.

The system uses the industrial tablet computer to install the INTOUCH HMI configuration software as the human–machine interface, which can ensure that the test bench starts operation according to the set test parameters. The monitoring acquisition system collects the operating data of each moving part, including torque, tension, speed, acceleration, and other data. The data acquisition cycle can be set, and the data chart curve can be automatically generated according to the recorded data.

The instrument panel graphics and text data list can display the following collected data: Left feeding inlet tension, right feeding inlet tension, conveying torque, left cutter torque, right cutter torque, left clamping conveying tension, right clamping conveying tension. Tension pressure data accuracy static: 1‰, tension pressure data dynamic static: 2‰, torque dynamic accuracy 2‰, each data can set the alarm value; if the data exceed the normal range, the system records and sends an alarm prompt. See Figure 17.

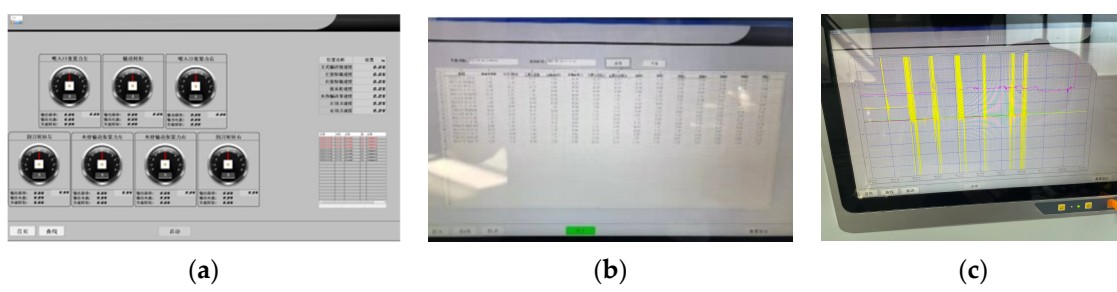

| (a) | (b) | (c) |

**Figure 17.** Data acquisition system: (**a**) test bench working parameters control interface; (**b**) data reports; (**c**) data curves.

According to the data obtained from the data acquisition system and the collection of cabbage in the aggregate box, the structural parameters and working parameters of each harvesting device in the harvesting system are adjusted in real time to reduce the damage rate of cabbage.

## 5. Single-Factor Test of Low-Loss Harvesting Test Platform

### 5.1. Test Object and Equipment

As shown in Figure 18, cabbage was selected as the test object, and the cabbage variety was "Chun xi". The expansion degree of cabbage is about 450–500 mm, the height of cabbage is about 140 mm, the width of cabbage is about 149 mm, the outer leaves are about 10–12, and the single cabbage quality range is 1.1–1.4 kg. By observing the working state of the pulling device, the reeling device, the clamping and conveying device, and the cutting device of the test platform. According to the characteristics of the cabbage harvested by the above device, the working frequency (0–50 Hz) of the servo motor driven by the above device is adjusted in real time.

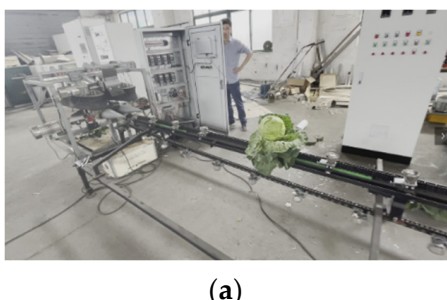 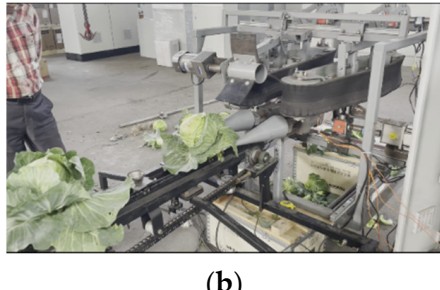

(**a**)                                                            (**b**)

**Figure 18.** Operation performance test: (**a**) conveying system of cabbage; (**b**) harvesting system of cabbage.

*5.2. Test Index and Test Results*

Since there are no related standards or regulations on mechanized cabbage harvesting. This experiment is based on GB/Z 26582-2011 Production Technical Practice for Cabbage [27] and JB/T 6276-2007 Testing Methods of Beet Harvesting Machinery [28] to select the qualified rate of harvesting as the standard for the harvesting test.

In the single-factor test affecting the qualified rate of cabbage harvest, other influencing factors should be fixed first. By changing one of the factors to obtain the variable relationship between the harvest pass rate and each test factor. Before the test, the machine is idle, and the test is carried out after the machine runs normally. Each group of tests was repeated 5 times, and 30 cabbages were tested each time. After the end of each group of experiments, the total number of test cabbages, the total number of successfully pulled cabbages, the total number of successfully clamped and transported cabbages, the total number of successfully cut cabbages, and the total number of qualified harvests were counted, respectively. Taking the qualified rate of cabbage harvest and the success rate of each link as the evaluation index, the initial level of each factor was set as follows: the rotation speed of the pulling roller was 100 r/min, the rotation speed of the conveyor belt was 160 r/min, and the rotation speed of the cutter head was 140 r/min.

(1) Single-factor test of extraction link: In the test, the speed of the fixed double disc cutter was 140 r/min, and the speed of the fixed flexible clamping conveyor belt was 160 r/min. The rotation speed of the pulling roller started from 60 r/min, and each comparison test increased by 20 r/min in turn. A total of 5 groups of tests were carried out, and the maximum rotation speed of the pulling roller test was 140 r/min.

(2) Single-factor test of flexible clamping conveying link: In the test, the rotation speed of the fixed pulling roller was 100 r/min, the rotation speed of the fixed double disc cutter was 140 r/min, and the rotation speed of the flexible clamping conveyor belt was increased from 80 r/min to 40 r/min each time. Five groups of tests were carried out, and the maximum rotation speed of the flexible clamping conveyor belt test was 240 r/min.

(3) Single-factor test of cutting link: In the test, the rotation speed of the fixed pulling roller was 100 r/min, the rotation speed of the fixed flexible clamping conveyor belt was 160 r/min, and the rotation speed of the cutter was increased from 100 r/min to 20 r/min each time. A total of 5 groups of tests were carried out. The maximum rotation speed of the double disc cutter test was 180 r/min.

The standard of mechanized harvesting of cabbage: At harvest, 2–3 outer leaves (rosette leaves) were retained to protect the leaf balls and ensure that the epidermis was clean and free of cracks.

Qualified rate of harvesting: The harvest qualified rate of cabbage was defined according to the production technical specifications of cabbage and the harvest quality requirements of stem and leaf vegetables: (1) The section of the cutting root must be smooth, and the two sections cannot be broken or cut out. (2) The cutting position should be 10–15 mm above the cabbage outer leaf, and the outer leaf should be cut off at the same time. (3) After

harvest, 2–3 outer leaves were retained to protect the leaf head. (4) No crack ball, extrusion damage, cutting damage, and so on caused by the harvesting operation occurred. The formula for the harvest-qualified rate is as follows:

$$N = \frac{N_1}{N_0} \times 100\% \tag{37}$$

where $N$ is the qualified rate of harvest, %; $N_1$ is the number of qualified cabbages harvested in a single test; and $N_0$ is the total number of cabbages harvested in a single test.

The test results of the pulling link, clamping and conveying link, and cutting link are shown in Tables 4–6.

**Table 4.** Analysis of single-factor test results of pulling roller speed.

| Rotating Speed of Pulling Roller (r/min) | Total Number of Cabbages Tested | Number of Damaged Cabbages | Number of Successfully Pulled Cabbages | Number of Qualified Cabbages Harvested | The Success Rate of Pulling (%) | Damage Rate (%) | Qualified Rate of Harvesting (%) |
|---|---|---|---|---|---|---|---|
| 60 | 30 | 2 | 28 | 26 | 90 | 6.7 | 86.7 |
| 80 | 30 | 1 | 26 | 25 | 86.7 | 3.3 | 83.3 |
| 100 | 30 | 1 | 29 | 28 | 96.7 | 3.3 | 93.3 |
| 120 | 30 | 3 | 29 | 26 | 93.3 | 10 | 86.7 |
| 140 | 30 | 4 | 29 | 25 | 96.7 | 13.3 | 83.3 |
| | | Average Value | | | 92.7 | 7.32 | 86.7 |

**Table 5.** Analysis of single-factor test results of clamping conveying speed.

| Rotation Speed of Clamping Conveyor Belt Speed (r/min) | Total Number of Cabbages Tested | Number of Damaged Cabbages | Number of Cabbages Successfully Transported by Clamping | Number of Qualified Cabbages Harvested | The Success Rate of Clamping Conveying | Damage Rate (%) | Qualified Rate of Harvesting (%) |
|---|---|---|---|---|---|---|---|
| 80 | 30 | 6 | 25 | 19 | 73.3 | 20 | 63.3 |
| 120 | 30 | 4 | 27 | 23 | 86.6 | 13.3 | 76.6 |
| 160 | 30 | 3 | 29 | 26 | 93.3 | 10 | 86.6 |
| 200 | 30 | 3 | 28 | 25 | 96.7 | 10 | 83.3 |
| 240 | 30 | 2 | 22 | 20 | 93.3 | 6.7 | 66.7 |
| | | Average Value | | | 88.6 | 12 | 75.3 |

**Table 6.** Analysis of single-factor test results of cutting speed.

| Rotation Speed of Double Disc Cutter (r/min) | Total Number of Cabbages Tested | Number of Damaged Cabbages | Number of Successfully Cut Cabbage | Number of Qualified Cabbages Harvested | Success Rate of Cutting | Damage Rate (%) | Qualified Rate of Harvesting (%) |
|---|---|---|---|---|---|---|---|
| 100 | 30 | 6 | 24 | 18 | 80 | 20 | 60 |
| 120 | 30 | 5 | 25 | 19 | 83.3 | 16.7 | 63.3 |
| 140 | 30 | 3 | 28 | 25 | 93.3 | 10 | 83.3 |
| 160 | 30 | 2 | 28 | 26 | 93.3 | 6.7 | 86.7 |
| 180 | 30 | 1 | 29 | 28 | 96.7 | 3.3 | 93.3 |
| | | Average Value | | | 89.3 | 11.34 | 77.3 |

In the single-factor test, the damage types of cabbage and the qualified cabbage are shown in Figure 19.

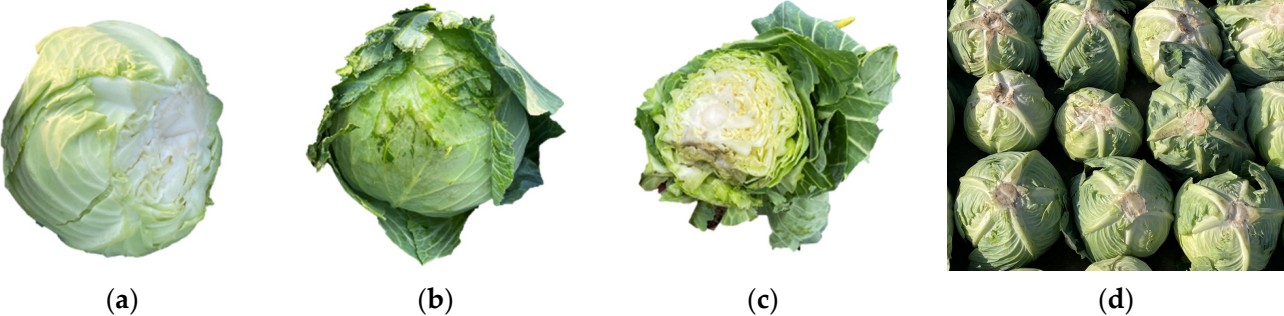

| (**a**) | (**b**) | (**c**) | (**d**) |

**Figure 19.** Damage types of cabbage: (**a**) broken cabbage; (**b**) scratched cabbage; (**c**) cabbage with cut loss; (**d**) harvest qualified cabbage.

## 6. Discussion

From the test results, it can be obtained that the average success rate of pulling of low-loss harvesting test platform was 92.7%, the average success rate of clamping and conveying was 88.6%, the average success rate of root cutting was 89.3%, the average qualified rate of harvesting in the pulling link was 86.7%, the average qualified rate of harvesting in the clamping and conveying link was 75.3%, and the average qualified rate of harvesting in the cutting link was 77.3%. The average damage rate of the pulling process was 7.32%, the average damage rate of the clamping and conveying link was 12%, and the average damage rate of the cutting process was 11.34%. During the test, the orderly feeding of cabbage can be realized by controlling the working frequency of the servo motor of the conveying system to avoid blockage and failure of operation performance. The pulling device and the reeling device can successfully pull the cabbage from the conveying bench, straighten up the cabbage to the feeding inlet of the clamping conveying device, and complete an effective harvest with the cutting device. Therefore, the test platform meets the technical requirements for the harvest of cabbage.

According to the test results, it can be inferred that when the pulling device is controlled at 80–120 r/min, the speed of the clamping and conveying device is controlled at 120–240 r/min, and the speed of the double disc cutter is controlled at 140–180 r/min, each link of the harvesting system has a high success rate. By observing the results of the aggregate box of each link, it can be seen that the damage rate of the pulling link in the harvesting system is low, while the damage rate of the clamping, conveying, and cutting links is high. Figure 16 shows the harvesting process and damage types of the low-loss harvesting test platform for cabbage. The reasons for the above phenomena are as follows: (1) The cabbage is not straightened when entering the clamping conveyor belt feeding inlet through the pulling device, which causes the root cutting position to shift. (2) The friction coefficient of the conveyor belt material is large, and the cabbage is scratched during the clamping and conveying process. (3) The too-fast or too-slow speed of the clamping conveyor belt makes the time of the cutter acting on the rhizome shorter in unit time, and the rhizome is not cut off or slipped. From the above causes of damage, it can be determined that the taper of the pulling roller, the material of the clamping conveyor belt, and the ratio of the clamping conveyor belt to the cutter head motion parameters are the key factors affecting the qualified rate of harvesting. Therefore, the experimental optimization will be carried out in the follow-up study in order to find out the critical conditions, the best equipment material, and the optimal working parameter combination for the damage of the cabbage in each link of the harvesting system and provide data support for the low-loss harvesting equipment for the cabbage.

## 7. Conclusions

(1) The basic physical characteristics and mechanical harvesting characteristics of cabbage, the representative of cabbage, were collected: the maximum shear force of the cabbage rhizome was 137.138 N, and the maximum crushing force was 1198.3 N.

(2) The low-loss harvesting process for cabbage was developed, and the structure of the prototype was determined. Combined with dynamics and kinematics analysis, the working parts of the pulling device, the reeling device, the clamping and conveying device, and the cutting device were designed and selected, and the above devices were integrated to design a low-loss harvesting test platform for the cabbage.

(3) The performance test of the low-loss harvesting test platform for cabbage shows that when the pulling device is controlled at 80–120 r/min, the speed of the clamping and conveying device is controlled at 120–240 r/min, and the speed of the double disc cutter is controlled at 140–180 r/min. Each link in the harvesting system has a high success rate. The success rate of pulling of low-loss harvesting test platform was 86.7–96.7%, the success rate of clamping and conveying was 73.3–96.7%, the success rate of root cutting was 80–96.7%, the qualified rate of harvesting in the pulling link was 76.7–93.3%, the qualified rate of harvesting in the clamping and conveying link was 63.3–86.7%, and the qualified rate of harvesting in the cutting link was 60–93.3%. The damage rate of the pulling process was 3.3–13.3%, the damage rate of the clamping and conveying link was 6.7–20%, and the damage rate of the cutting process was 3.3–20%.

**Author Contributions:** Conceptualization, W.T. and J.Z.; methodology, W.T. and J.Z.; software, W.T. and Z.S.; validation, W.T., J.Z. and Z.S.; formal analysis, W.T. and X.N.; investigation, W.T.; resources, W.T.; data curation, G.C.; writing—original draft preparation, W.T.; writing—review and editing, W.T.; visualization, Z.S.; supervision, G.C.; project administration, J.Z., G.C. and Z.S. All authors have read and agreed to the published version of the manuscript.

**Funding:** This research was funded by the National Natural Science Foundation of China (Grant No. 52205273), Jiangsu Agriculture Science and Technology Innovation Fund (Grant No. CX(22)2020), Key R&D Program of Shandong Province in China (Grant No. 2022CXGC010612), Innovation Project of Chinese Academy of Agricultural Sciences (Grant No. CAAS-ASTIP-NIAM).

**Institutional Review Board Statement:** Not applicable.

**Data Availability Statement:** The datasets used and/or analyzed during the current study are available from the corresponding author on reasonable request.

**Conflicts of Interest:** The authors declare no conflict of interest.

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
