# Peer review of "Design and Experiment of a Low-Loss Harvesting Test Platform for Cabbage"

_agriculture, doi:10.3390/agriculture13061204_

Round 1

Reviewer 1 Report

The full text is clear in thinking and standardized in writing. The cabbage low-loss harvesting test platform developed has certain innovation and has certain guidance for actual production. As far as the content of text writing is concerned, there are still some details that need attention :

1.The units of the full text should be unified, such as r/min in the summary and r·min-1 in table 1.

2.Line 187, whether to consider which key parts the device consists of.

3,Line 227, whether the trajectory integration of the reel is needed, please consider.

4.In addition to cabbage, whether the test platform developed in this paper is suitable for other knotty leafy vegetables, I think this can increase the universality of the test platform, please consider.

5.The squeezing pressure of the loading conveyor is much smaller than the maximum crushing force, is there a phenomenon that the squeezing pressure is too small to be successfully loaded, and will the appropriate increase in squeezing pressure increase the probability of successful loading, please add explanation.

very well

Reviewer 2 Report

The paper completed the collection of the basic physical characteristics of cabbage and designed a low-loss harvesting test platform for cabbage. The overall structure of the paper is complete, and the topic selection has certain creativity and high academic research reference value.

As far as the writing content of this article is concerned, it is suggested to adjust the following contents:

1. Pay attention to the academic language expression of the whole article, and further improve the article format. Such as the accurate expression of the unit, the writing format needs to be further standardized.

2. Line 140, whether the physical characteristics of the cabbage machine should be placed before the whole machine mechanism and working principle.

3. line 235, the force analysis of cabbage on the pulling roller was not in place, no specific conclusions were drawn, and the selection of subsequent influencing factors did not play a theoretical support role. It is recommended to add detailed force analysis instructions.

4. Some references were cited incorrectly, e.g. 24, 25, etc.

Reviewer 3 Report

It is increasingly widespread attention for how to improve the operational performance of cabbage harvester and reduce the harvesting damage. As selecting kale for the test object, the manuscript design a low-loss test platform for harvesting a typical representative of knotty leafy cabbage. It is hoped to solve the urgent problem of the high damage rate in the process of cabbages' harvesting. This manuscript is excellent in both innovation and practicality.

However, there are some aspects throughout the manuscript needed to be improved:

First, in view of the importance of success rate and damage rate for the low-loss harvesting test platform, it is appropriate aspect to appear in pairs for the two parameters in the abstract section.

Second, in my opinion, all units should be unified in the full text. For example, in the line 139 of manuscript, Table 1. The main technical parameters, the writing of r-min-1 and m / s should be unified.

Third, there's one detail that makes me wonder: on account of the coefficient of variation for the dimensionless counting method, the location of this parameter in Table 3 makes the data ambiguous. At the same time, what is the coefficient of variation result of the pulling force, 0.18% or 18%? The variation coefficient of 18% means that the dispersion degree of extraction force measured is large. the possibility of error in the data is higher. It is very necessary and important for confirming and modifying the data.

Fourth, It is mentioned in the manuscript that the design of the extraction roll can meet the theoretical extraction force of the previous test. however, I don't have the details of the structural analysis process. Adding and revising it carefully, I think, is very necessary.

The last one, I don't think it suitable for giving a data scope of success rate and other evaluation parameters for the harvesting test platform. The parameters should be considered and modified. If not, the reader will be extremely confused.

Although there are so many aspects need to be modified and improved, it does not obscure the innovation and practicality of this manuscript. It was my suggestion to publish it after a slight revision.

The Quality of English Language is that minor editing of English language should be required.
